# Short Sleep Duration Was Associated with Increased Regional Body Fat in US Adults: The NHANES from 2011 to 2018

**DOI:** 10.3390/nu14142840

**Published:** 2022-07-11

**Authors:** Chong Xu, Song Zhao, Shikai Yu, Jiamin Tang, Han Zhang, Bei Xu, Yawei Xu, Yi Zhang

**Affiliations:** 1Department of Cardiology, Shanghai Tenth People’s Hospital, Tongji University School of Medicine, Shanghai 200072, China; xu1528763821@163.com (C.X.); zhaosqq@163.com (S.Z.); shikaiyu@yahoo.com (S.Y.); logfund@163.com (J.T.); 2Department of Nuclear Medicine, Shanghai Tenth People’s Hospital, Tongji University School of Medicine, Shanghai 200072, China; 17317557160@163.com; 3Department of Endocrinology and Metabolism, Shanghai Tenth People’s Hospital, Tongji University School of Medicine, Shanghai 200072, China

**Keywords:** sleep duration, body fat, NHANES

## Abstract

Background: The relationship between sleep duration and different regional fat is unclear. We aimed to investigate the association between sleep duration and different regional fat mass (FM) among a population of US adults. Methods: 9413 participants were included from the National Health and Nutrition Examination Survey (NHANES), from 2011 to 2018. The sleep duration was divided into short sleep (<7 h/day), normal sleep (7–9 h/day) and long sleep (>9 h/day). Different regional FM was measured by dual-energy X-ray absorptiometry, including trunk FM, arms FM and legs FM. Fat mass index (FMI) was obtained by dividing FM (kg) by the square of body height (m^2^). Multiple linear regression was used to evaluate the relationship between sleep duration and regional FMI. Results: The mean sleep duration was 7.1 ± 1.5 h/day. After adjusting for socio-demographic, lifestyle information, comorbid diseases and medications, short sleepers had higher trunk FMI (β = 0.134, 95% confidence interval (CI): 0.051–0.216, *p* = 0.001), arms FMI (β = 0.038, 95% CI: 0.016–0.06, *p* < 0.001) and legs FMI (β = 0.101, 95% CI: 0.044–0.158, *p* < 0.001) compared to normal sleepers, whereas no significant difference was found in long sleepers. The similar results were also observed in men, while short sleepers only had higher arms FM in women (all *p* < 0.01). In addition, compared to normal sleepers, short sleepers had higher arms FMI and legs FMI in the obese group (all *p* < 0.05), while no relationship was observed in non-obese group. Conclusions: Short sleep duration, but not long sleep duration, was independently related to the increased different regional body fat in US adults, especially in men and those with obesity.

## 1. Introduction

Obesity has become a global epidemic in recent decades. Obesity can lead to a higher risk of non-communicable diseases such as cardiovascular diseases, diabetes and cancer, resulting in a serious socio-economic burden [1,2,3]. Therefore, it is necessary to take appropriate measures to prevent and treat obesity. Lifestyles aimed at reducing energy intake and increasing calorie consumption can delay the progression of obesity, such as optimal diet and exercise [4]. In addition, adequate sleep can maintain immune system homeostasis and have a positive feedback effect on metabolism [5,6]. However, more than one-third of American adults sleep less than 7 h [7]. Sleep deficiency can lead to metabolic disorders [8,9]. Therefore, there is an urgent need for a multidimensional assessment of the relationship between sleep duration and obesity, which can help provide guidance for weight management.

Previous studies have examined the link between sleep duration and obesity. KIM et al. found that adults with short sleep (<5 h/day) were more likely to develop obesity defined by body mass index (BMI) in the Korean Health and Nutrition Survey study [10]. In a large population of wearable sensor users, participants with higher BMI were prone to shorter sleep duration [11]. Besides, Bacaro et al. conducted a meta-analysis and found that short sleep duration was related to an increased risk of obesity in adults, while long sleep duration was not [12]. To sum up the above, most studies have primarily used BMI to assess obesity, with limited ability to distinguish between fat mass (FM) and fat-free mass, which failed to accurately assess the role of sleep and adiposity [13]. Further, Tan et al. discovered that both habitual short and long sleep were correlated with higher FM by bio-impedance analysis in a Swedish cohort study [14]. However, regional variability exists in adipose tissue and the relationship between different regional fat and sleep duration has rarely been reported in adults [15,16]. To fill this gap, we aimed to explore the relationship between weekday sleep duration and different regional (trunk, arms and legs) fat in a multi-ethnic cohort of US adults.

## 2. Materials and Methods

### 2.1. Study Design and Participants

The National Health and Nutrition Examination Survey (NHANES) is a large cross-sectional study designed to assess the health and nutritional status of children and adults across the United States. The survey began in 1960s and consist mainly of interviews and physical examinations. In 1999, the survey has been conducted on a 2-year cycle. About 5000 participants are selected for data collection each year [17]. Data from 2011–2012 to 2017–2018 were applied to this analysis. A total of 39,156 participants were enrolled in the study, excluding those younger than 18 years of age (*n* = 15,331), missing sleep duration data (*n* = 103), missing dual-energy X-ray absorptiometry (DXA) data (*n* = 12,635) and missing covariate data (*n* = 1674), resulting in the inclusion of 9413 participants (Figure 1). Notably, 12,635 participants with missing DXA data included 7632 participants aged > 59 years who did not undergo DXA scans. The Institutional Review Board of the National Center for Health Statistics approved the study protocol, and all participants signed consent forms.

### 2.2. Definition of Regional Fat Mass

Whole-body DXA scans were performed on participants aged 8–59 years by professional operators, excluding pregnancy; self-reported history of radiographic contrast (barium) use within the past 7 days; self-reported weight more than 450 pounds or height over 6’5”. Hologic Discovery model A densitometers (Hologic, Inc., Bedford, MA, USA) were used for scanning. All DXA scans needed to undergo quality control and analysis, resulting in measurements of soft tissue and bone in various parts of the body. The present analysis included total FM, left and right arm FM, left and right leg FM and trunk FM. The arms and legs FM were defined as the sum of both limbs FM respectively. Fat mass Index (FMI) was obtained by dividing FM (kg) by height squared (m^2^).

### 2.3. Definition of Sleep Duration

Sleep duration was self-report by participants. Sleep duration was defined by the following question: “How much sleep do you usually get at night on weekdays or workdays?” According to the expert consensus of the American Academy of Sleep Medicine and the Sleep Research Society [18,19], sleep duration was divided into short sleep duration (<7 h/day), normal sleep duration (7–9 h/day) and long sleep duration (>9 h/day).

### 2.4. Covariate Assessment

A standardized household interview questionnaire was used to collect socio-demographic and lifestyle information, including age, gender, race, education, marital status, smoking, alcohol status, sedentary time, physical activity, work status, comorbid diseases and medications. Race was divided into five groups: Mexican American, Other Hispanic, non-Hispanic white, non-Hispanic black and other race. Education was categorized into under high school, high school or equivalent and at least college. Marital status was classified as married or living with partner, widowed, divorced or separated and never married. According to the questions: “Have you smoked at least 100 cigarettes in your entire life?” and “Do you now smoke cigarettes?”, participants were divided into current smokers, former smokers and non-smokers. Alcohol status was created according to the question: “Had at least 12 alcohol drinks a year?” Sedentary time (min/day) was defined from a question: “How much time do you usually spend sitting on a typical day?” The total physical activity time (TPAT) was calculated based on two questions: “How much time do you spend doing vigorous-intensity activities at work on a typical day?” and “How much time do you spend doing moderate-intensity activities at work on a typical day?” Then, TPAT was divided into <150, 150–300, 300–450, ≥450 min/day. Work status is sorted according to the question: “Type of work done last week?” Diabetes treatment was defined by two questions: “Are you now taking diabetic pills to lower blood sugar?” and “Are you now taking insulin?” Hyperlipidemia and lipid-lowering therapy are separately defined according to the following questions: “Have you ever been told by a doctor that your blood cholesterol level was high?”, “Have you ever been told to take prescribed medicine?” and “Are you now following this advice to take prescribed medicine?”

Professionals measured participants’ standing body height, body weight and waist circumference at the Mobile Examination Center. Body mass index (kg/m^2^) was calculated as body weight (kg) divided by height squared (m^2^). Obesity was defined as a BMI ≥ 30 kg/m^2^. Fasting blood glucose, glycated hemoglobin, total cholesterol and high-density lipoproteins were measured by standardized methods. Diabetes was defined as a previous diagnosis of diabetes or current diabetes treatment, or glycosylated hemoglobin ≥ 6.5% or fasting glucose > 7 mmol/L [20].

### 2.5. Statistical Analysis

Baseline information was divided into three groups based on sleep duration. Continuous variables were expressed as mean ± standard deviation and categorical variables were expressed as absolute numbers (percentage). The histogram was used to describe the overall distribution of sleep duration. Differences in total and regional FMI were examined between sleep groups according to gender and BMI subgroups. Differences between groups were analyzed by one-way ANOVA and chi-square test. Two-way comparisons were performed by Bonferroni post hoc analysis and chi-square splitting when necessary. Multiple linear regression was used to evaluate the relationship between sleep duration (categorical variable) and total and regional FMI, expressed as β and 95% confidence intervals (CI). Model 1 was adjusted for age, sex and race. Model 2 was adjusted for Model 1, plus education, marital status, TPTA, sedentary time and work status. Model 3 was adjusted for Model 2, plus diabetes, diabetes treatment, use of stain, smoke status, alcohol status, total cholesterol and high-density lipoprotein. Since individuals with higher BMIs had shorter sleep duration, which was more likely to impact regional fat, subgroup analysis was performed by BMI, adjusted for Model 3. All statistical tests were two-sided, and *p* < 0.05 was defined as statistically sigificant difference. Statistical analyses were performed using SAS (version 9.4, SAS Institute, Cary, NC, USA) and R (version 4.1.1; R Foundation, Auckland, New Zealand).

## 3. Results

### 3.1. Participant Characteristics

The participants in this study included 4851 men and 4562 women. The mean age was 37.61 ± 12.2 years. The average BMI was 28.5 ± 6.6 kg/m^2^ and the average waist circumference was 96 ± 16.2 cm. The mean total, trunks, arms, legs FMI were 9.61 ± 4.3, 4.59 ± 2.3, 1.18 ± 0.6 and 3.42 ± 1.6 kg/m^2^, respectively. The average sleep duration was 7.1 ± 1.5 h/day (Appendix A). Table 1 analyzes the demographic sociology, lifestyle, comorbid diseases, and medication use of different sleepers. Compared to normal sleepers, short sleepers had higher regional FMI in total and gender subgroups (all *p* < 0.01) (Appendix A).

### 3.2. Association of Sleep Duration with Different Regional FMI in Whole Population

In the minimal adjusted model (Model 1), compared to normal sleepers, short sleepers had higher total FMI (β = 0.384, 95% CI: 0.217–0.550, *p* < 0.001), trunk FMI (β = 0.204, 95% CI: 0.114–0.296, *p* < 0.001), arms FMI (β = 0.054, 95% CI: 0.03–0.078, *p* < 0.001) and legs FMI (β = 0.122, 95% CI: 0.063–0.181, *p* < 0.001). There was no significant difference in regional FMI between long sleepers and normal sleepers (all *p* < 0.05) (Table 2). Immediately after, in the multivariate model (Model 3), short sleep duration was still related to higher total FMI (β = 0.275, 95% CI: 0.121–0.430, *p* < 0.001), trunk FMI (β = 0.134, 95% CI: 0.051–0.216, *p* = 0.001), arms FMI (β = 0.038, 95% CI: 0.016–0.060, *p* < 0.001) and legs FMI (β = 0.101, 95% CI: 0.044–0.158, *p* < 0.001), while this relationship was not observed in long sleepers (Table 2).

### 3.3. Differences by Gender and BMI Strata

As shown in Table 3 and Figure 2, after adjustment for covariates, compared to the reference group, short sleepers and increased trunk FMI (β = 0.126, 95% CI: 0.029–0.223, *p* = 0.011), arms FMI (β = 0.031, 95% CI: 0.008–0.055, *p* = 0.009), legs FMI (β = 0.108, 95% CI: 0.047–0.171, *p* < 0.001) were significantly correlated in men. However, in women, short sleep duration was only associated with elevated arms FMI (β = 0.042, 95% CI: 0.005–0.080, *p* = 0.027). Furthermore, in the obese group (Table 4), short sleepers had higher arms and legs FMI compared to normal sleepers (all *p* < 0.05). This relationship was not observed in the non-obese group (all *p* > 0.05).

## 4. Discussion

Based on a multiracial cohort of the American adults, there are several important findings. First, we discovered that short sleep duration was associated with higher trunk, arms and legs fat, whereas this relationship was not found in the long sleep group compared to the normal sleep group. Secondly, a similar relationship was identified in men, but sleep deficiency was only related to increased arms fat in women. Finally, we found a relationship between short sleep duration and regional fat only in the obese group.

As a biological behavior throughout life—sleep has a profound impact on health [21]. Several studies have revealed the relationship between sleep and body composition. In a cohort of German adults, Rahe et al. evaluated sleep quality by the Pittsburgh Sleep Questionnaire Index (PSQI) and found that participants with poor sleep quality had more adiposity [22]. Another Korean cohort also reported an association between self-reported sleep deprivation and elevated body fat [10]. However, in a cross-sectional study enrolled middle-aged adults, higher PSQI was not related to increased whole-body FM [23]. The variation in the above studies may be attributed to differences in population, sleep assessment and body fat measurement methods. We found that short sleep duration was still associated with increased total FMI after adjustment for covariate factors. Possible mechanisms are that poor sleep can diminish leptin secretion and insulin sensitivity, in turn leading to the imbalance in energy intake and expenditure, which results in fat accumulation [24,25]. In addition, a meta-analysis discovered that short sleep duration was associated with an increased risk of diabetes, so increased fat might play a mediating role [26,27]. The aforementioned studies mainly dealt with the relationship between sleep and total body fat, while studies on the relationship between sleep and regional fat are scarce. Sweatt et al. observed that poor sleep quality was associated with increased trunk fat and visceral fat [23]. In another Mendelian randomized study, the investigators discovered a two-way causal relationship between snoring and right leg and trunk FM [28]. Our study found that short sleepers had more fat in the trunk, upper limbs and lower limbs compared to normal sleepers. This may reveal a homogeneity of short sleep duration on regional fat gain. The relationship can be further confirmed by prospective and experimental studies.

Prior studies have shown that women have more body fat and gender differences exist in fat distribution [29]. However, fewer studies have explored the gender differences between sleep duration and body fat. KIM et al. identified an association between sleep shortage and elevated total FM regardless of gender [22]. Nonetheless, in a cohort of Chinese adults, nocturnal sleep deprivation and increased FMI were associated only in men [30]. In the present study, sleep deficiency was associated with elevated total, trunk, arms and legs FMI in men, but only with total, arms FMI in women. The distribution of sleep duration by gender was further analyzed. It was found that women slept more at all ages (Appendix A). This might be one of the reasons why regional fat gain was not significant in women. In addition, female participants were middle-aged and estrogen may slow down the increase in fat, attenuating the effect of sleep duration [31].

Interestingly, our study uncovered an association between short sleep duration and regional adiposity only in the obese group. Differences in sleep quantity, quality and sleep-related breathing disorders exist among people of various body sizes [32]. In a longitudinal cohort of adults wearing activity monitoring devices, Jaiswal et al. found that higher BMI was associated with objectively shorter sleep duration [11]. We also found that participants with higher BMI had shorter sleep duration (Appendix A). Previous studies reported individuals with higher BMI are more likely to develop obstructive sleep apnea (OSA) [33,34]. Patients with OSA are more susceptible to sleep fragmentation, resulting in shorter sleep duration [35,36]. This may partially explain the variability in the relationship between short sleep duration and regional adiposity across BMI groups. Subsequent studies are warranted to help understand the causal relationship between regional fat, OSA and sleep duration.

This study has several advantages. Body fat was measured by the gold standard DXA. Furthermore, we explored the relationship between sleep duration and different regions of adiposity in a multi-ethnic adult population. Yet there are still some limitations. Sleep duration is self-reported and participants were prone to recall bias. Sleep duration assessed by the gold standard including polygraphy or actigraphy can be applied for further studies. Secondly, the current analysis only involved weekday sleep duration. However, weekend sleep duration was longer among most adults [37], so an additional effect of weekends sleep duration on regional adiposity could not be excluded. Furthermore, multidimensional analysis of sleep phenotypes was lacking. Thirdly, DXA screening was performed only in young and middle-aged adults, so it was inappropriate to extend the results to the elderly. Finally, the study was cross-sectional and the causality could not be determined.

## 5. Conclusions

Short sleep duration, rather than long sleep duration, was independently associated with regional fat in US adults, especially in men. Besides, short sleep duration was only related to elevated regional fat in the obese group.

## Figures and Tables

**Figure 1 nutrients-14-02840-f001:**
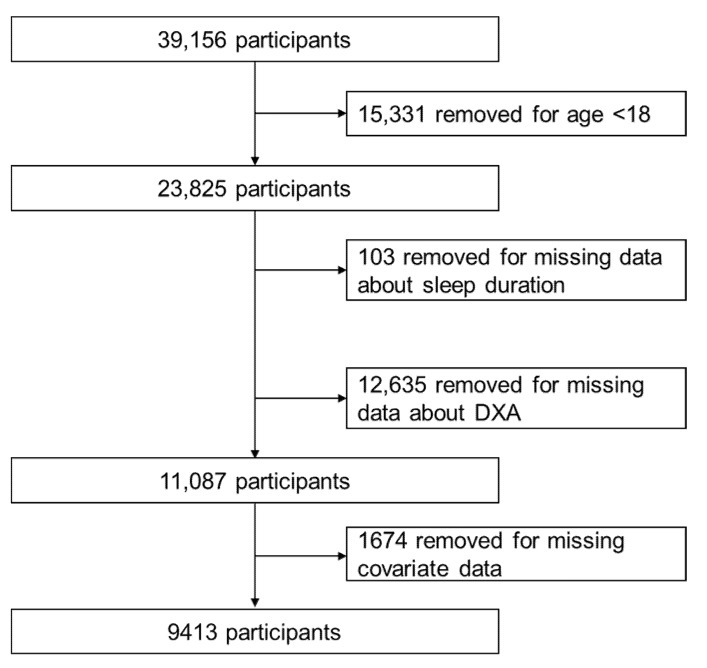
Flowchart of participants inclusion from NHANES 2011–2018.

**Figure 2 nutrients-14-02840-f002:**
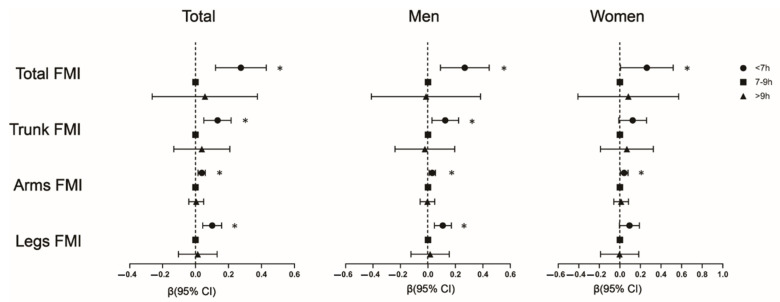
Multivariable linear regression of sleep duration groups with different regional fat mass. Adjusted confounders include age, race, sex (only in total), education level, marital status, physical activity, sedentary activity, work, diabetes, diabetes treatment, use of stain, smoke, alcohol, total cholesterol and low-density lipoprotein. Compared to the normal sleep group, the short sleep group was more likely to have increased different regional FMI in total and in men, while short sleepers only have higher total FMI and arms FMI in women. No significant difference was found in long sleepers. * (Higher than normal sleep duration groups) represents *p* < 0.05. FMI: fat mass index.

**Table 1 nutrients-14-02840-t001:** Characteristics of participants grouped by short, normal, and long sleep duration.

	Sleep Duration (h/day)			*P* Value
	<7 (*n* = 3126)	7–9 (*n* = 5771)	>9 (*n* = 516)	
Age (years)	39.00 ± 11.9	37.31 ± 12.2	32.46 ± 12.7	<0.001
Sex, *n* (%)				<0.001
Men	1703 (54.48%)	2916 (50.53%)	232 (44.96%)	
Women	1423 (45.52%)	2855 (49.47%)	284 (55.04%)	
Race, *n* (%)				<0.001
Mexican American	425 (13.60%)	961 (16.65%)	96 (18.60%)	
Other Hispanic	331 (10.59%)	588 (10.19%)	55 (10.66%)	
Non-Hispanic White	1029 (32.92%)	2117 (36.68%)	167 (32.36%)	
Non-Hispanic Black	839 (26.84%)	924 (16.01%)	107 (20.74%)	
Other Race	502 (16.06%)	1181 (20.46%)	91 (17.64%)	
Education, *n* (%)				<0.001
Under high school	570 (18.23%)	1035 (17.93%)	135 (26.16%)	
High school or equivalent	751 (24.02%)	1369 (23.72%)	188 (36.43%)	
At least College	1805 (57.74%)	3367 (58.34%)	193 (37.40%)	
Marital status, *n* (%)				<0.001
Married or living with partner	1752 (56.05%)	3333 (57.75%)	200 (38.76)	
Widowed	49 (1.57%)	58 (1.01%)	2 (0.39%)	
Divorced or separated	434 (13.88%)	573 (9.93%)	50 (9.69%)	
Never married	891 (28.50%)	1807 (31.31%)	264 (51.16%)	
Current smoking, *n* (%)	776 (24.82%)	1065 (18.45%)	148 (28.68%)	<0.001
Alcohol consumption, *n* (%)	2442 (78.12%)	4392 (76.10%)	359 (69.57%)	<0.001
Sedentary activity (min/day)	376.4 ± 211	379.1 ± 199.7	347.1 ± 197.8	0.003
Total physical activity (min/day)	141.2 ± 220.8	110 ± 195.4	106.9 ± 182	<0.001
Current working, *n* (%)	2368 (75.75%)	4225 (73.21%)	244 (47.29%)	<0.001
Diabetes, *n* (%)	329 (10.52%)	437 (7.57%)	43 (8.33%)	<0.001
Hyperlipidemia, *n* (%)	774 (24.83%)	1249 (21.67%)	92 (17.86%)	<0.001
Hypoglycemic therapy, *n* (%)	223 (7.45%)	281 (4.87%)	26 (5.04%)	<0.001
Lipid-lowering therapy, *n* (%)	282 (9.02%)	376 (6.52%)	34 (6.59%)	<0.001
Waist circumference (cm)	98.1 ± 16.8	95.2 ± 15.7	93.5 ± 16.9	<0.001
BMI (kg/m^2^)	29.2 ± 6.9	28.1 ± 6.4	27.6 ± 6.6	<0.001
Total cholesterol (mg/dL)	189.3 ± 40.7	188.9 ± 40.2	181.3 ± 38.1	<0.001
High-density lipoprotein (md/dL)	51.4 ± 15.3	52.4 ± 15.1	53.0 ± 15.4	0.006

BMI: body mass index.

**Table 2 nutrients-14-02840-t002:** Association between different regional fat mass and sleep duration groups in the whole population.

Duration	Model 1		Model 2		Model 3	
(h/day)	β (95% CI)	*p* Value	β (95% CI)	*p* Value	β (95% CI)	*p* Value
Total FMI (kg/m^2^)						
<7	0.384 (0.217, 0.550)	<0.001	0.381 (0.214, 0.547)	<0.001	0.275 (0.121, 0.430)	<0.001
7–9	reference		reference		reference	
>9	−0.022 (−0.365, 0.320)	0.898	0.006 (−0.339, 0.350)	0.974	0.057 (−0.263, 0.376)	0.729
Trunk FMI (kg/m^2^)						
<7	0.204 (0.114, 0.296)	<0.001	0.204 (0.113, 0.294)	<0.001	0.134 (0.051, 0.216)	0.001
7–9	reference		reference		reference	
>9	0.003 (−0.184, 0.190)	0.973	0.012 (−0.176, 0.200)	0.898	0.038 (−0.132, 0.208)	0.661
Arms FMI (kg/m^2^)						
<7	0.054 (0.030, 0.078)	<0.001	0.054 (0.030, 0.077)	<0.001	0.038 (0.016, 0.060)	<0.001
7–9	reference		reference		reference	
>9	0.002 (−0.047, 0.050)	0.940	−0.001 (−0.050, 0.048)	0.975	0.004 (−0.041, 0.049)	0.862
Legs FMI (kg/m^2^)						
<7	0.122 (0.063, 0.181)	<0.001	0.120 (0.061, 0.178)	<0.001	0.101 (0.044, 0.158)	<0.001
7–9	reference		reference		reference	
>9	−0.028 (−0.149, 0.093)	0.650	−0.006 (−0.127, 0.115)	0.922	0.013 (−0.104, 0.131)	0.824

Model 1: adjusted for age, sex and race. Model 2: adjusted for age, sex, race, education level, marital status, physical activity, sedentary activity and work. Model 3: adjusted for the covariates in Model 2, diabetes, diabetes treatment, use of stain, smoke, alcohol, total cholesterol and high-density lipoprotein. FMI: fat mass index.

**Table 3 nutrients-14-02840-t003:** Association between different regional fat mass and sleep duration groups in gender groups.

Duration	Men		Women	
(h/day)	β (95% CI)	*P* Value	β (95% CI)	*P* Value
Total FMI (kg/m^2^)				
<7	0.268 (0.091, 0.446)	0.003	0.262 (0.006, 0.518)	0.045
7–9	reference		reference	
>9	−0.014 (−0.412, 0.384)	0.945	0.082 (−0.407, 0.572)	0.742
Trunk FMI (kg/m^2^)				
<7	0.126 (0.029, 0.223)	0.011	0.125 (−0.010, 0.259)	0.070
7–9	reference		reference	
>9	−0.022 (−0.240, 0.195)	0.841	0.068 (−0.189, 0.325)	0.603
Arms FMI (kg/m^2^)				
<7	0.031 (0.008, 0.055)	0.009	0.042 (0.005, 0.080)	0.027
7–9	reference		reference	
>9	−0.004 (−0.057, 0.049)	0.878	0.012 (−0.059, 0.084)	0.734
Legs FMI (kg/m^2^)				
<7	0.108 (0.047, 0.171)	<0.001	0.093 (−0.004, 0.190)	0.061
7–9	reference		reference	
>9	0.016 (−0.124, 0.155)	0.826	−0.003 (−0.188, 0.183)	0.979

Model: adjusted for age, race, education level, marital status, physical activity, sedentary activity, work, diabetes, diabetes treatment, use of stain, smoke, alcohol, total cholesterol and high-density lipoprotein. FMI: fat mass index.

**Table 4 nutrients-14-02840-t004:** Association between different regional fat mass and sleep duration groups in obese groups.

Duration	BMI ≥ 30		BMI < 30	
(h/day)	β (95% CI)	*P* Value	β (95% CI)	*P* Value
Total FMI (kg/m^2^)				
<7	0.263 (0.026, 0.500)	0.030	0.060 (−0.039, 0.160)	0.234
7–9	reference		reference	
>9	−0.176 (−0.697, 0.345)	0.507	0.098 (−0.101, 0.297)	0.336
Trunk FMI (kg/m^2^)				
<7	0.114 (−0.012, 0.240)	0.077	0.026 (−0.028, 0.080)	0.350
7–9	reference		reference	
>9	−0.144 (−0.422, 0.134)	0.309	0.080 (−0.029, 0.188)	0.151
Arms FMI (kg/m^2^)				
<7	0.047 (0.009, 0.084)	0.014	0.004 (−0.010, 0.017)	0.605
7–9	reference		reference	
>9	−0.033 (−0.115, 0.049)	0.429	0.014 (−0.013, 0.041)	0.299
Legs FMI (kg/m^2^)				
<7	0.099 (0.003, 0.196)	0.044	0.031 (−0.009, 0.071)	0.131
7–9	reference		reference	
>9	0.005 (−0.208, 0.218)	0.966	0.001 (−0.079, 0.082)	0.972

Model: adjusted for age, sex, race, education level, marital status, physical activity, sedentary activity, work, diabetes, diabetes treatment, use of stain, smoke, alcohol, total cholesterol and high-density lipoprotein. FMI: fat mass index

## Data Availability

All data are available at https://wwwn.cdc.gov/nchs/nhanes (accessed on 1 April 2022).

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
