# Peer review of "Short Sleep Duration Was Associated with Increased Regional Body Fat in US Adults: The NHANES from 2011 to 2018"

_nutrients, 2022, doi:10.3390/nu14142840_

Round 1

Reviewer 1 Report

The manuscript titled “Sleep deprivation, but not over-sleep, was associated with increased
regional body fat in US adults: the NHANES from 2011 to 2018”. It addresses the issue of whether habitual sleep deprivation (e.g. due to life-styles) or sleep extension is associated to different regional body fat. Participants were 9,413 from the National Health and Nutrition Examination Survey (NHANES), that had enrolled partecipants aged  18-85 years from 2011 to 2018. Participants were divided into 3 groups on the bases of their reported sleep duration: short sleepers (<7h/day), normal sleepers (7-9h/day) and long sleepers (>9h/day). The following regional fat mass (FM) were measured by absorptiometry: trunk FM, arms FM and legs FM. Results evidenced that short sleepers had higher FM indexes in all regions considered, whereas no significant difference was found in long sleepers. Gender and nutritional group (groups obtained by BMI)  differences in this relationship were also found: They were all significantly correlated in men while in women, sleep deprivation was only associated with elevated arms FM. Moreover, in the obese group, short sleepers had higher arms and legs FM index compared to normal sleepers while the relationship was not observed in the non-obese group.

The study was well conducted and the sample size is impressive. However the following pittfalls should be outlined and addressed by authors before publication.

Introduction:

The starting point of the study, and thus the theoretical rational that informed it, is summarized int the following statement: “Previous studies have examined the link between sleep duration and obesity, but with mixed results”. The authors cite several studies in support of this inconsistency but they seem to be completely unaware that there are many reviews and meta-analyses (e.g. rifs) addressing this issue that allow very general conclusions about the fact that only short sleep and not long sleep is consistently associated to higher body fat. Thus the rational cannot be the inconsistency of previous results

Method:

Sleep duration was self-reported and measured through the following single item: “How much sleep do you usually get at night on weekdays or work-days?”. Although many studies have used self-report data instead of what is considered the gold standard for assessing sleep duration (polygraphy or at least actigraphy). This should be addressed in the discussion.

Reviewer 2 Report

By utilizing data from NHANES, Xu et al. investigated the association between self-reported sleep duration and fat mass distribution in 9413 American adults. The study is well conducted and DXA-determined fat mass in a large population is a strength. However, there are some concerns over the current version of the manuscript.

1.     Over half of the initial participants were excluded due to missing DXA outcomes, was DXA data missing at random? Must be clarified in the manuscript.

2.     Abstract: in addition to the P values, please give the adjusted mean differences (or beta value and CI) for the main findings of the study. Statistical methods and confounding factors should also be mentioned in the abstract.

3.     Please mention in the discussion that only weekday situation was accounted in the assessment of sleep duration. In working-age adults a 1-2 h gap in sleep duration between weekdays and weekends are common, this has not been considered in the present study.

4.     Sleep deprivation is not equivalent to short sleep duration, please be consistent and keep using the latter form throughout the paper, same for long sleep duration.

5.     Figure 2 and Figure 3 can be reported as supplements.

6.     Why stratify the data by BMI 30 (obesity)? Is there a significant interaction between BMI and sleep duration regarding total FM and regional FM?

7.     Short sleep duration is associated with an increased risk of type 2 diabetes in general population. Findings from the study adds new evidence for the candidate mechanism (adipose tissue distribution) associating sleep with diabetes. Please add relevant discussion (see Wang et al. 2021 J Cachexia Sarcopenia Muscle PMID 34595832).
